# Viral Metagenomics in the Clinical Realm: Lessons Learned from a Swiss-Wide Ring Trial

**DOI:** 10.3390/genes10090655

**Published:** 2019-08-28

**Authors:** Thomas Junier, Michael Huber, Stefan Schmutz, Verena Kufner, Osvaldo Zagordi, Stefan Neuenschwander, Alban Ramette, Jakub Kubacki, Claudia Bachofen, Weihong Qi, Florian Laubscher, Samuel Cordey, Laurent Kaiser, Christian Beuret, Valérie Barbié, Jacques Fellay, Aitana Lebrand

**Affiliations:** 1Global Health Institute, Swiss Federal Institute of Technology (ETH Lausanne) & SIB Swiss Institute of Bioinformatics, 1015 Lausanne, Switzerland; 2Institute of Medical Virology, University of Zurich, 8057 Zurich, Switzerland; 3Institute for Infectious Diseases, University of Bern, 3001 Bern, Switzerland; 4Institute of Virology, VetSuisse Faculty, University of Zurich, 8057 Zurich, Switzerland; 5Functional Genomics Center Zurich, Swiss Federal Institute of Technology (ETH Zurich) & University of Zurich, 8057 Zurich, Switzerland; 6Laboratory of Virology, University Hospitals of Geneva, 1205 Geneva, Switzerland; University of Geneva Medical School, 1206 Geneva, Switzerland; 7Biology Department, Spiez Laboratory, 3700 Spiez, Switzerland; 8Clinical Bioinformatics, SIB Swiss Institute of Bioinformatics, 1202 Geneva, Switzerland; 9Precision Medicine Unit, Lausanne University Hospital and University of Lausanne, 1010 Lausanne, Switzerland

**Keywords:** viral metagenomics, ring trial, external quality assessment, EQA, quality control

## Abstract

Shotgun metagenomics using next generation sequencing (NGS) is a promising technique to analyze both DNA and RNA microbial material from patient samples. Mostly used in a research setting, it is now increasingly being used in the clinical realm as well, notably to support diagnosis of viral infections, thereby calling for quality control and the implementation of ring trials (RT) to benchmark pipelines and ensure comparable results. The Swiss NGS clinical virology community therefore decided to conduct a RT in 2018, in order to benchmark current metagenomic workflows used at Swiss clinical virology laboratories, and thereby contribute to the definition of common best practices. The RT consisted of two parts (increments), in order to disentangle the variability arising from the experimental compared to the bioinformatics parts of the laboratory pipeline. In addition, the RT was also designed to assess the impact of databases compared to bioinformatics algorithms on the final results, by asking participants to perform the bioinformatics analysis with a common database, in addition to using their own in-house database. Five laboratories participated in the RT (seven pipelines were tested). We observed that the algorithms had a stronger impact on the overall performance than the choice of the reference database. Our results also suggest that differences in sample preparation can lead to significant differences in the performance, and that laboratories should aim for at least 5–10 Mio reads per sample and use depth of coverage in addition to other interpretation metrics such as the percent of coverage. Performance was generally lower when increasing the number of viruses per sample. The lessons learned from this pilot study will be useful for the development of larger-scale RTs to serve as regular quality control tests for laboratories performing NGS analyses of viruses in a clinical setting.

## 1. Introduction

Clinical microbiology laboratories have been increasingly investing in next generation sequencing (NGS) in recent years. Clinical applications include, for example, outbreak investigation, viral typing and drug resistance analysis, pathogen (shotgun) metagenomics, and microbiome (amplicon-based) metagenomics.

In particular, the shotgun metagenomics using NGS is a promising technique to analyze both the DNA and RNA microbial material from patient samples (e.g., [1,2,3,4,5,6,7,8,9]). Mostly used in a research setting, it is now increasingly being used in the clinical realm as well, notably to support the diagnosis of viral infections in unknown aetiologies, systemic infections, or for virus discovery.

Clinical metagenomics is a complex workflow including several critical steps from sample preparation up to reporting of pathogens. In particular, it heavily relies on bioinformatics for data processing, analysis and interpretation. Efforts have been made to benchmark bioinformatics tools for the detection of bacteria from patient samples [10], for internal laboratory validation of clinical metagenomics assays for pathogen detection in cerebrospinal fluid [1] and to establish guidelines for the validation of clinical metagenomics assays, also using spiked samples and artificial data [11].

In Switzerland, the Swiss Institute of Bioinformatics (SIB) leads and coordinates the field of bioinformatics. In 2016, SIB launched a nation-wide working group (WG) on the NGS Microbes Typing and Characterization, with the aim of harmonizing NGS practices within Swiss clinical microbiology/virology laboratories, notably regarding bioinformatics. The WG currently comprises clinical and bioinformatics experts from all Swiss university hospitals and their associated clinical microbiology/virology labs, cantonal hospitals, the Swiss Federal Institute for NBC-Protection (Spiez laboratory), as well as research groups from Swiss academic institutions, having met eight times face-to-face since September 2016.

Following an initial detailed survey on NGS practices at Swiss hospitals and virology labs, highlighting the variety of methodologies and software used across the country, the WG suggested that SIB organizes a viral metagenomics ring trial, as a first step towards benchmarking current workflows used in clinical settings, defining common best practices, and importantly, paving the way towards quality-controlled routine implementation of NGS in clinical virology laboratories through participation in external quality assessment (EQA) programs.

The viral metagenomics ring trial was therefore designed to be a quality control test for pathogen identification from viral metagenomics data. The main objectives were to benchmark existing workflows currently used at Swiss hospitals and clinical virology laboratories, quantify differences among the participating laboratories, identify where improvements and training might be needed, and foster increased quality and the development of common best practices for NGS viral metagenomics. We present here the results from the viral metagenomics ring trial (RT) that was run over 2018.

## 2. Materials and Methods

### 2.1. Design of the Ring Trial

The ring trial consisted of two increments, selected along the NGS data analysis pipeline running from the sample preparation up to reporting (Figure 1). The first increment started from biological material, to compare results across the whole workflow. Then, the second increment removed the laboratory variability by starting directly from the sequencer’s raw reads, and assessed the extent of variability stemming from the bioinformatics part alone. This design aimed at disentangling the variability arising from the wet lab compared to the bioinformatics parts of the pipeline.

In addition, the ring trial was also designed to assess the impact of reference sequences databases compared to the bioinformatics algorithms on the final results. Thus, at each increment, participants performed the bioinformatics analysis twice, once with a SIB-provided common database (see Section 2.2), and once using their own in-house database (if applicable).

#### 2.1.1. Increment 1

Participants received:
Questionnaire on methodology used (cf. Section 2.6).Five samples consisting of the human blood plasma spiked with known viruses (cf. Section 2.3).SIB common database (cf. Section 2.2).

Participants returned:
Filled questionnaire on the methodology that was used for each pipeline (done only once when first joining the RT, either at increment 1 or 2).Raw reads from the sequencer (FASTQ), before further pre-processing.Classified reads (e.g., BAM, BGA if aligning reads; or e.g., TSV if using a k-mer-based approach).The identified viruses with associated metrics (virus metrics file): Virus|database_ID|Number_of_reads|percent_coverage|genome_sizeA technical lab report containing the results interpretation and methodological details.

Note: We ended up not using the classified reads for increment 1 (BAM, BGA files), as the ground truth for each read was unknown, and participants were already asked to compute summary statistics including read counts in the virus metrics file.

#### 2.1.2. Increment 2

Participants received:Thirteen FASTQ datasets labeled with a random number, consisting of five samples from increment 1 sequenced by some of the participants, and eight in-silico generated FASTQ datasets (cf. Section 2.4).SIB common database (cf. Section 2.2).

Participants returned the same output as for increment 1, except for the raw reads (FASTQ). In addition, the virus metrics table was updated to contain two additional columns: (i) Report (“1” if the virus would be reported to clinicians; “0” otherwise), and (ii) comment.

### 2.2. Swiss Institute of Bioinformatics (SIB) Common Database

All sequences annotated as complete viral genomes (around 70,000) were downloaded from the GenBank (query “VRL[Division] AND ‘complete genome’ [ALL]”). These sequences were then subjected to screening (based on entry annotations) for the following criteria: (i) The sequence must be of a complete genome; (ii) the virus must have a vertebrate host; (iii) the sequence must not contain unknown nucleotides. Entries were then sorted by clade, that is, viruses known to the species level were grouped by genus, viruses known only to the genus level were grouped by family, etc. Clades containing fewer than two entries were dropped, and no clade was allowed to exceed 20 entries (20 entries were drawn at random from clades with 21 or more; excess entries were discarded). Entries were then split into a test and a training set by attributing every fifth entry to the test set and all the others to the training set. Only the training set was provided to participants. The test set was used to generate the in-silico datasets of increment 2, as explained in Section 2.4 below.

The common database, consisting of the training set, was provided in three versions: One with reference viral sequences only, one supplemented with non-target/contaminant sequences (bacteria, fungi, bacteriophages, etc.), and one supplemented with human sequences.

In order to make sure that the common database could be seamlessly integrated into the participants’ analysis pipelines, the headers in the FASTA file were generated so as to comply with the conditions indicated by participants on the registration form.

The SIB-provided common database is available as Appendix A. Note, however, that during the course of the ring trial, several reference viral databases have become available to the scientific community as well (e.g., Virosaurus, a curated database available on request to philippe.lemercier@sib.swiss (personal communication); and the Reference Viral Database (RVSB) [12]).

### 2.3. Viral Samples for Increment 1

Five different samples were prepared for increment 1.

Three samples consisted of unfiltered healthy donor plasma (Blutspende Zürich, Schlieren, Switzerland) spiked with four viruses obtained from the in-house cell culture supernatant (1:1, 1:10, 1:100 fold serial dilutions in plasma):Human betaherpesvirus 5Human mastadenovirus BEnterovirus CInfluenza A virus

In order to control the dilutions, total nucleic acid was extracted with the NucliSENS eMAG (bioMérieux, Marcy l’Etoile, France) followed by qPCR in triplicates with specific primers. The Ct values for all four viruses were in the range of 24.9 to 26.3 for the 1:1 dilution. 1 mL aliquots were stored at −20 °C until shipment on dry ice.

In addition, a viral multiplex control [13] from the National Institute for Biological Standards and Control (NIBSC) was prepared according to the protocol. The reagent contains a dilution of the following viruses in human plasma: AdV-2, BKV, HHV-5 (CMV), HHV-4 (EBV), HSV-1, HSV-2, HHV-6A, HHV-6B, JCV, B19, and VZV. The lyophilized content of six vials was resuspended, pooled, mixed and 1 mL aliquots were stored at −20 °C until shipment on dry ice. The Ct value for HHV-4 was 28.9, for the other ten viruses it was in the range of 31.6 to 33.8. 

Finally, a NIBSC negative control [14] was prepared by pooling the content of two vials of 4 mL, mixed and 1 mL aliquots were stored at −20 °C until shipment on dry ice.

The resulting five identically-looking samples were labeled with a number from one to five (Table 1) and shipped frozen to the contact person of each participating laboratory, with the instruction to keep samples at −20 °C until further processing.

### 2.4. FASTQ Datasets for Increment 2

In increment 2, in order to cancel differences due to laboratory procedures and sequencing technology, all participants received the same set of sequencing reads in FASTQ format, which they subjected to the bioinformatics part of their pipeline as if they had been actual reads from their sequencing step.

We provided 13 FASTQ datasets to participants, five of which were obtained from real samples, and eight consisted of in-silico generated reads. All samples were relabled with a random number from one to 13 (Table 1, column “Label”).

#### 2.4.1. Real Sequencing Reads

We selected five FASTQ datasets among the data produced by the participants of increment 1. These five datasets consisted of the 1:10 and 1:100 fold serial dilutions of the spiked samples, the NIBSC negative control, and twice the NIBSC viral multiplex control (sequenced by two different laboratories). 

Since pipeline H was the only one to sequence DNA and RNA separately, we selected the five datasets among the data provided by the remaining three sequencing centers (i.e., pipelines A, E, I), to make sure that every participating laboratory could use their existing workflow without having to integrate results from two separate analyses. Table 1 (column “Pipeline”) shows the pipeline from where we picked each FASTQ dataset. The settings (single-end compared to the paired-end; read length) are also indicated in Table 1 (column “Settings”).

Before sending out these datasets, FASTQ headers were anonymized, the sequencing quality was checked with FastQC, and we compared the output results that had been provided for that dataset in increment 1 compared to what was expected, to make sure that the participating laboratory had not swapped samples by mistake.

#### 2.4.2. Artificial Sequencing Reads

The reads were produced with art_illumina [15] using source sequences in FASTA format from the test set of the common database, that is, sequences that were not provided to the participants (cf. Section 2.2). These included sequences from viral origin, but also sequences from contaminants (bacterial, human and other known contaminants). The actual origin of the reads was not communicated to the participants and was masked in the FASTQ headers.

We devised two challenges, with four datasets each, representing in total eight datasets. Participants were not informed that some datasets contained the same viruses, and datasets were labeled with a random number (Table 1, column “Label”).

The first challenge, consisting of datasets “II”, contained a preset number of sequences from one strain of each of the following viruses, in addition to contaminants as explained above:Human mastadenovirus AHuman coronavirus HKU1Severe acute respiratory syndrome-related coronavirus (SARS)Influenza B virusHuman respirovirus 1 (HPIV-1)Human rubulavirus 4 (HPIV-4)Human orthopneumovirus (Human respiratory syncytial virus)Human metapneumovirusNorwalk virusRotavirus AHepacivirus C (HCV)

We then performed a 1:40 fold serial dilution, a 1:40 fold serial dilution increasing the mutation rate in art_illumina, and a 1:400 fold serial dilution to obtain four datasets (Table 2).

The second challenge, consisting of datasets “III”, contained a preset number of sequences from one strain of each of the following viruses, in addition to contaminants as explained above:Human betaherpesvirus 5 (CMV)Human alphaherpesvirus 1 (HSV-1)Human betaherpesvirus 6A (HHV-6A)Parechovirus AHuman alphaherpesvirus 3 (VZV)JC virusBK virus

We then performed a 1:10 fold serial dilution, a 1:10 fold serial dilution increasing the mutation rate in art_illumina, and a 1:100 fold serial dilution to obtain four datasets (Table 2).

Note that dilution datasets were re-generated from the test set FASTA sequences and were not simply a subsampling of the corresponding 1:1 FASTQ dataset. The scripts used to generate the in-silico datasets are available in Appendix A.

In addition, each of these eight datasets was generated for several combinations of settings (single (one) compared to paired-end (two) × read length):1 × 100 bp1 × 150 bp1 × 250 bp2 × 100 bp2 × 150 bp2 × 250 bp

The aim was that participants could choose the settings corresponding to their actual workflow. However, in order to ensure fairness among participants, we decided to normalize the number of reads per virus to the read length, as having more reads is certainly an advantage for virus detection and identification, but that comes at a cost, resulting in a necessary trade-off to enable a routine implementation in a clinical setting. Thus, when generating the datasets, we made sure that the provided viral information, measured as the total number of bases of viral origin (i.e., number of reads of viral origin x average read length), was identical in all these settings. As an example, the dataset generated for 1 × 150 bp contained only 67% of reads of viral origin as compared to the dataset generated for 1 × 100 bp (as 100/150 = 67%). In addition, the fractions of contaminants of human and non-human origin were invariably fixed to 54% and 45% of the total number of reads, respectively, leaving 1% of reads of viral origin in each dataset. The exact number of reads in each dataset can be found in Appendix A.

The datasets provided in increment 2 are available in Appendix A.

### 2.5. Implementation of the Ring Trial

The ring trial was organized over 2018. Participants had three months to perform increment 1 (February to May 2018), and 2.5 months to perform increment 2 (July to September 2018).

Each laboratory had the possibility to participate with one or several pipelines. By pipeline we refer to a specific combination of laboratory and bioinformatics workflows. Participants were asked to use identical settings for a given pipeline in all the increments and when using the in-house or SIB-provided databases.

Four laboratories participated in increment 1, for a total of seven pipelines; five laboratories participated in increment 2, for a total of seven pipelines as well (one drop out and one joining for increment 2 only). The methodologies used in the eight pipelines are briefly described in Table 3.

Each pipeline was assigned a capital letter, and samples were numbered. When naming the output files, participants were asked to prefix all filenames using the convention (pipeline_letter) + (sample_number), e.g., A4.bam for the BAM file of sample 4 obtained with pipeline A. In addition, for the in-silico datasets, participants had also to specify the sequencing parameters that they had chosen from the list that we were providing (e.g., B4_2 × 100.bam, referring to paired-end reads of 100 bp).

Participants had to answer a short questionnaire on the methodology that they used (see Section 2.6). The questionnaire was pipeline-specific and password-protected by a pipeline-specific token. At the end of the questionnaire, participants had direct links to upload small files to sync.com (password-protected). For larger data files (FASTQ, BAM), participants could also submit them using a SWITCHfilesender voucher (50 Gb), SWITCH being the protected cloud from the Swiss academic community.

### 2.6. Questionnaire

The questionnaire consisted of 24 questions covering:StorageSample preparation (enrichment)DNA/RNA extraction, quantification, quality assessmentLibrary preparationSequencingBioinformatics (reads pre-processing, methodology)

The list of questions is available in Appendix A.

### 2.7. Analysis of Results

#### 2.7.1. Pre-Processing of the Metrics File

All the virus metrics files were converted to the csv format, and typos in the headers were manually corrected to match the expected headers.

For increment 1, participants were not asked to mention in the virus metrics file whether they would report the virus or not. Having a “report” column was however extremely handy to automatically analyse the results, and we therefore decided to manually add a “report” column into the metrics file returned by the participants, relying on their reports to transcribe this information. If the NCBI accession number was mentioned in the report, we simply set the corresponding virus in the metrics file as “1” in the “report” column, since the metrics file contained a database_ID field with NCBI accession numbers. Reports mentioning “Uninterpretable” or “Cannot be excluded” were considered as not reported. Unfortunately, some reports did not mention at all the NCBI accession numbers but only a virus name. We, therefore, had to manually go over all the viruses in the metrics file to identify which viruses were actually the ones reported in the final report. For increment 2, this manual step was removed by directly asking participants to add a “report” column to their metrics file.

#### 2.7.2. Virus Identification

The expected viruses in each sample were known. Since viruses can have multiple names and since we only required a species-level resolution to assess performance, we defined regular expressions representing the various names and subtypes of each of the expected viruses (Appendix A).

We used the NCBI accession number provided for each reported virus in the metrics file (i.e., “report” column set to “1”) to retrieve the virus full name using the NCBI E-utilities API. The full virus name was then mapped against the regular expressions representing the expected positives (Appendix A), in order to determine the number of true positives (TP), false negatives (FN) and false positives (FP) identified by each pipeline, and thus the resulting sensitivity, precision and F1-score. Note that we did not use the virus name provided by participants in the metrics file, as for the in-house databases, this virus name sometimes corresponded to a higher-level resolution (e.g., genus), as the participating laboratory found it more convenient to report viruses present in the sample in more generic terms to clinicians, thereby not reflecting the actual capacity of the laboratory to narrow down the virus at the species level.

#### 2.7.3. Analysis of Results

For the spiked samples (i.e., non-negative), performance was measured as the F1-score, which is the harmonic mean of sensitivity and precision. Sensitivity, also known as recall or true positive rate, is measured as TP/(TP + FN), where TP are the true positives and FN are the false negatives. Thus, sensitivity reflects the fraction of true positives among all the expected viruses to be found. On the other hand, precision, also known as positive predictive value, is measured as TP/(TP + FP), and therefore reflects the fraction of true positives among all the predicted positives. The analysis was carried out with Ruby and R [R Core Team 2019] scripts.

False negatives were mapped to family-level resolution, in order to be able to compare the false positives identified across pipelines (Appendix A).

In each sample, for each of the reported viruses (i.e., report = 1 in the virus metrics file), we estimated the depth of coverage using the following formula:depth = number of reads mapping to that virus * average read length/virus genome size(1)

For each sample, we then calculated a mean depth of coverage by taking the harmonic mean of the depths across all the reported viruses in that sample.

All the analysis scripts are available in Dataset S4.

## 3. Results

### 3.1. Great Variability in Overall Performance

In increment 1, participants received five biological samples, one of which was a negative sample. All the pipelines differed in some way, sometimes only at the bioinformatics level (cf. Table 3). Indeed, for the seven pipelines, there were actually five sets of submitted FASTQ datasets from four sequencing centers, as pipelines E, F and J were based on a common FASTQ dataset.

We observed a significant variability in performance across pipelines. As shown in Figure 2, pipelines H and I had a sensitivity and precision of 100% on all three dilutions of the spiked samples, resulting in an F-score equal to one. They were followed by E, F, J, A and B. Thus, on the spiked samples of increment 1, pipelines based on mapping achieved a better performance than those based on k-mers (cf. Table 3). We observe, however, that performance on the NIBSC multiplex viral control was in general poorer, although virus concentrations were in a similar range than for the 1:100 dilution of the spiked sample. For this sample, mapping-based methods did not systematically outperform the k-mers-based methods. The poorer performance on the viral multiplex may be explained by the large number of different viruses and the fact that this sample resembled less a real clinical sample.

In increment 2, four samples were the same as in increment 1. We represent in Figure 3 the results for those samples in increment 1 compared to increment 2. Despite participants now starting from the same FASTQ dataset, we still observed a great variability in performance for the spiked samples in increment 2. In addition, it is noticeable that while pipeline E was able to replicate its results when given its own datasets (with anonymized headers), pipeline I achieved different performances with the same FASTQ dataset at increment 1 compared to 2.

In addition, we generated eight in-silico datasets (described in Materials and Methods), for which we plotted the performance across pipelines in Figure 4. Contrary to increment 1, no pipelines stood out as top performers. Datasets II were particularly challenging, both at the sensitivity and precision levels. On the other hand, participants achieved generally good precision for datasets III, but had very low sensitivity (not shown), explaining that the overall performance measured as the F1-score was generally better for datasets III than II (Figure 4). Interestingly, the higher mutation rate (samples four and one, triangles in Figure 4) did not impact the overall performance and generally had F1-scores comparable to their corresponding dilution (squares in Figure 4). Serial dilutions did, however, result in decreased performance.

All the identified viruses in samples of increments 1 and 2 are available as Appendix A.

### 3.2. Depth of Coverage per Virus Is Correlated with Performance

In order to better understand what drives performance, we plotted the performance as a function of the mean depth per reported virus (Figure 2; cf. Materials and Methods). Interestingly, in increment 1, we observe a positive correlation between the mean depth per virus and the performance, with a plateau around 10× (Figure 2a).

In order to see if the mean depth was directly linked to the sequencing depth, we plotted in Appendix A the performance as a function of the total number of sequenced reads. Interestingly, we observe that pipeline H sequenced 50–100 times more reads than the rest of the pipelines, which indeed translates as well into ~2 orders of magnitude higher depth per virus as compared to the other pipelines (Figure 2a), but without an increase in performance versus pipeline I. Thus, for the types of samples provided in increment 1, it appears that combining a performing bioinformatics pipeline with sequencing 5–10 × 10⁶ reads per sample may be sufficient for virus identification. Other sample types or applications (e.g., stool samples, viruses discovery) may however require other sequencing settings, e.g., deeper sequencing.

Rationalising costs for routine use of metagenomics is an important factor to take into account. In addition, reduced turnaround times are also critical to enable clinical impact of this technology and foster its adoption in the clinic. We plotted in Appendix A the performance as a function of turnaround time from sample preparation to the report, for the samples of increment 1. Interestingly, we observe no correlation at all, with pipeline I achieving great performance in very short turnaround times (33 h = 1.4 days) as compared to the other pipelines.

### 3.3. Impact of Sample Preparation

In increment 2, participants received two FASTQ datasets originating from the same sample, but prepared and sequenced by two different laboratories (NIBSC_multiplex_a prepared and sequenced by laboratory I; NIBSC_multiplex_b prepared and sequenced by laboratory E). Although our RT only included one such duplicate, the results show that participants consistently performed much worse with sample NIBSC_multiplex_b than with sample NIBSC_multiplex_a (Figure 3). 

Interestingly, the average performance of all pipelines on NIBSC_multiplex_a was similar to that of pipeline I at increment 1, while the average performance of all pipelines on NIBSC_multiplex_b was similar to that of pipeline E at increment 1. Altogether, these results strongly suggest that sample preparation can have a very large impact on the overall pipeline performance. A possible explanation for the observed differences in performance when looking at the samples preparation of pipelines I and E could be the nuclease pre-treatment included in pipeline E (Table 3). In the future RT, it would be interesting to further test the impact of sample preparation by having several of these duplicates (or even triplicates for a given sample).

We note, however, that for NIBSC_multiplex_a, for which participants generally achieved a good performance, the F1-score still varied notably across pipelines (Figure 3), mostly due to variability in precision (not shown). Thus, bioinformatics methodology also has a strong impact on the overall performance.

In order to further investigate how much sample preparation may be driving overall performance, we also compared the participants performance at increment 2 compared to increment 1 for the spiked samples (Figure 3). The increment 2 Spiked_1-10 dataset was provided by laboratory E, and the increment 2 Spiked_1-100 dataset was provided by laboratory I, with a respective performance on those samples at increment 1 of ~75% and 100%. Interestingly here, we still observe great variability in the F1 performance at increment 2 across all the pipelines, which is driven by a high variability in precision (not shown). This suggests that the bioinformatics methodology also has a strong impact on overall performance, in particular the capability of avoiding calling false positives. Taken together, these results suggest that sequencing should be deep enough to achieve a 10× coverage of viruses, and that this is necessary but not sufficient for achieving high sensitivity and precision, as we still observe great variability of performance across different bioinformatics workflows. 

As we mentioned, most of the variability that we observed in this RT often arose from differences in precision rather than sensitivity. Appendix A shows the family of the false positive viruses identified by each pipeline in the negative sample of increment 1 (NIBSC_negative). Apart from pipeline H, which did not report any virus in this negative sample, we observe that all the remaining pipelines did report at least two viruses from various families. We wondered to what extent these false positives were due to laboratory contaminants [16] (which may be corrected for by using appropriate negative controls), experimental sequencing carry-over or index hopping, or bioinformatics mapping procedures (note that all participants used the same SIB-provided database of viruses, but may have decided not to use the contaminants database that was also provided). Figure 5 shows the results obtained for the NIBSC_negative sample in increment 2, where this time the FASTQ dataset was the same for all the participants (sequenced in increment 1 by pipeline A (cf. Table 1)). Interestingly, if the false positives were all stemming from laboratory or experimental contaminants, we would expect all pipelines to call the same false positives. This is indeed the case for two families of viruses (Polyomaviridae and Adenoviridae), consistently reported by all the pipelines as present in the negative sample. All the remaining families of false positive viruses were, however, reported by only some of the pipelines, suggesting that these may arise from the bioinformatics workflow per se. In particular, we note that pipeline H only reported the two families of viruses reported by everyone and no other false positive. Thus, even in the absence of a negative control to run alongside its samples (which likely explains the absence of false positives reported in increment 1), pipeline H also seems to have been tuned to achieve low numbers of false positives.

### 3.4. Impact of Database Quality and Size

One of the aims of the RT was to test for the importance of the reference sequences database that is used to map/classify the reads taxonomically compared to the mapping/classification algorithm. Thus, for both increments, SIB provided a common database that contained whole genome sequences of viruses and known “contaminants” (bacteria, fungi, host, etc.). Participants were asked to perform the analyses with the common database (“SIB”), in addition to using their “in-house” database. Figure 6 shows the F1-score obtained by the five pipelines that participated in increment 1 and used both the SIB and an in-house database. These results show no significant difference in performance for any given pipeline using the SIB compared to the in-house database (Mann-Whitney U test), suggesting that in our RT, it is the mapping/classification algorithm that is driving the overall performance, rather than the reference sequences database that is used along the algorithm. We note that most of the F1-score signal was driven by a differential performance in precision across the pipelines, rather than sensitivity (Appendix A).

The size of the used database in terms of the number of viral sequences did not have an impact on overall performance either (not shown).

## 4. Discussion

The SIB working group on “NGS Microbes Typing and Characterization” and the ring trial that we implemented aimed at harmonizing NGS practices in clinical viral metagenomics. Harmonizing means here for different clinical laboratories to achieve comparable results of high quality, irrespective of the methodology chosen at the experimental or bioinformatics levels, which may result in part from internal constrains as well (e.g., existing laboratory processes for nucleic acid extraction to be re-used as much as possible for metagenomics as well). Our results have highlighted various aspects that can impact the overall performance at the experimental, databases and bioinformatics levels, offering insights to each participating laboratory into processes where they may further improve their workflows. In order for others to also benefit from this study and test their own workflows, we have published with this manuscript the database and all the datasets that were generated (Appendix A).

### 4.1. Lessons Learned and Recommendations

We learned several lessons while implementing this pilot viral metagenomics ring trial. In particular, we want to emphasize the need to use NCBI accession numbers together with regular expressions (Appendix A) and NCBI Taxonomy in order to correctly assign each reported virus to the taxonomic resolution of interest (species-level in our case). Indeed, in clinical practice, laboratories use different names for the same virus, and we observed that they sometimes reported the virus with a more common name despite having identified it at the species level, to make their reports more reader-friendly for clinicians. When establishing such ring trials, it is therefore important to clearly ask participants to report viruses at the most detailed resolution level possible, and provide them with means to report this (e.g., NCBI Taxonomy, or NCBI accession number). This is of particular importance since viral taxonomy is not consistently defined, and thus asking for virus identification may mean different taxonomic names and levels for different laboratories depending on the virus. The reported accession number will be useful to retrieve the virus full name, before asking if it belongs to the expected list of viruses at the chosen resolution level. In this sense, it is also essential that the list of allowed names for each of the expected viruses at the chosen resolution level is made available after such proficiency tests for participants review (cf. Appendix A in our ring trial). For increment 2, we also decided to ask participants to directly indicate in the metrics file which viruses they would report to clinicians, as the produced reports did not always mention NCBI accession numbers.

Our ring trial was based on spiked plasma samples. Our results in Figure 2a suggest that in this context, a depth per virus of about 10×, as achieved by pipeline I, may be a rational target for high quality output results. Achieving a sequencing depth of 10× per reported virus can, however, be difficult and not always feasible, especially in samples other than plasma having a higher biomass and for low replicating viruses. Thus, others have suggested to use instead≥3 non-overlapping reads in distinct genomic regions as a criterion for reporting viruses in cerebrospinal fluid samples [1]. Ultimately, the target depth of coverage might likely depend on the envisioned application and type of sample. This may also have consequences on what can be expected in terms of sensitivity and precision, and what sample processing steps, bioinformatics tools and interpretation criteria may need to be fine-tuned accordingly to compensate for e.g., the lower depth of coverage per virus in some applications. As a result, the performance criteria notably in an accredited setting may end up being application-specific to some extent, or require, e.g., additional complementary validation tests to confirm the metagenomics results, as discussed in [3] for the identification of viral encephalitis, meningoencephalitis and meningitis. In the end, it is important to bear in mind that clinical metagenomics assays will likely not be used alone or as systematic first-line tests, but rather in combination with other diagnostic tests that all together will let clinicians take the best decisions for the patient.

In this ring trial, participants generated their results with a common, SIB-provided database, in addition to using their own in-house database of reference viral sequences. One of the aims was to assess the impact of the database compared to the algorithms on performance. Interestingly, we observed that the database that we provided enabled participants to get as good results as they would get using their own in-house database (Figure 6). One reason for this finding may be that participating laboratories have hired bioinformaticians and dedicate resources in (semi-automatically/automatically) refining their internal databases and do not simply download all NCBI viruses, likely achieving already good quality databases similar to the one we provided. Thus, in our settings (i.e., rather common viruses and Swiss clinical context), it appears that the impact of the database was rather low on the overall performance. It is however unclear how generalizable this finding may be, given the limited number of tested viruses in our samples and the small number of participating laboratories. Since we did not test several databases of varying qualities, we would still recommend taking advantage of the expert curated databases that are being published (cf. Material and Methods and references therein) to optimize the speed of bioinformatics workflow, decrease the false positive calls in general (cf. Appendix A), and also increase sensitivity for less sequenced viruses. In general, for the implementation of future ring trials, we find that providing a common database is not necessary, as long as the in-house databases of participants include NCBI accession numbers to enable the tracking of each virus to the desired taxonomic level.

Our ring trial was designed in such a way that the ground truth was always known in both increments. This was very practical to be able to assess precision and sensitivity. In increment 1, this came with the caveat that samples did not necessarily reproduce clinical conditions, although we tried as much as possible to replicate real conditions, notably by using plasma from human donors. On the other hand, the NIBSC viral multiplex control clearly did not come close to a real clinical scenario, and this may explain the observed much lower performance on these samples due to the high number of viruses in the sample. This highlights how much the context of the sample (and patient) are also essential in guiding the interpretation of viral metagenomics results.

We also found it very informative to use the same samples in increment 2 as in increment 1 (also with a duplicate), but this time providing a common FASTQ dataset to all the participants (picked from one of the participants from increment 1, but anonymizing the headers). Surprisingly, for the laboratories from whom we had picked the FASTQ datasets, we noticed that the performance achieved at increment 1 was not always replicated in increment 2, despite starting with identical FASTQ files. This notably highlights the importance of human interpretation at the end of the bioinformatics workflow. Indeed, while some participants indicated having clear interpretation criteria (e.g., minimum absolute number of reads mapping to a virus, minimum number of non-overlapping reads, minimum percentage of coverage) in addition to manual inspection, several participants also mentioned performing only a manual inspection. Interestingly, depth of coverage was not mentioned as one of the reporting criteria, although we observed in increment 1 (Figure 2a) that a depth of coverage of 10× and beyond was associated with top performance. We could however not reproduce this finding in increment 2 (Figure 2b), perhaps highlighting the fact that workflows have been optimized as a whole (i.e., from sample to report), and not in separate blocks (e.g., laboratory procedures only, or bioinformatics workflow only). This finding perhaps questions the utility of performing ring trials assessing only one part of the workflow, by e.g., starting from common FASTQ datasets. From another perspective, this also highlights that participants should be cautious when analyzing data that do not originate from their own laboratory. Ultimately, human interpretation very likely has a substantial impact on overall performance. In this sense, we also suggest that the clinical viral community should share more interpretation and reporting practices, in order to further harmonize their practices and increase overall knowledge with more expertise sharing. The impact of human review and the observation that one pipeline did not reproduce its findings starting from an identical FASTQ dataset also calls for individual laboratories more clearly defining internal standard interpretation criteria, which would help reduce the impact of the differential human review and likely be essential in an accredited context to ensure results reproducibility over time.

Due to the design of the ring trial, participants of the ring trial did not have the possibility to use negative controls that were representative of the samples being tested, since they lacked this information. In increment 1, pipelines E, F, H and J nevertheless used internal controls alongside the provided samples (Appendix A). Pipeline H used a negative control (water submitted to the whole NGS process), whereas pipelines E, F and J used a well characterized sample as an internal control (weak positive for one virus). Of these pipelines, however, only pipeline H correctly identified the negative sample as being negative, thus being able to correctly deplete real contaminants from its data, while not adding type-I errors due to wrong mapping/classification in the bioinformatics step. Pipelines E, F and J however still reported viruses in the negative sample, suggesting these may arise mostly from type-I errors during the bioinformatics step. In increment 2 (Figure 5), we also observed that it is likely laboratory contaminants and type-I errors that affect the participants’ false positive rate. In this sense, always running a negative control alongside is certainly very good practice to deplete laboratory contaminants, but not sufficient to avoid false positives calls in the bioinformatics step. 

### 4.2. From Pilot Studies to Accredited External Quality Assessment (EQA)

Nowadays, as NGS for viral diagnostics is on the brink of being used in routine, laboratories currently perform viral metagenomics in a non-accredited context. The increasing number of laboratories implementing NGS for shotgun metagenomics calls for quality management and the implementation of ring trials to benchmark different pipelines and ensure comparable results across sites in terms of performance, within clinically-realistic costs and turnaround times. Ultimately, participating in such ring trials will be essential for laboratories to get accredited in using shotgun metagenomics for viral diagnostics.

As of today, few ring trials (or proficiency tests) have been put in place with a particular focus on viral (or pathogen) metagenomics. We can mention (i) the initiative led by COMPARE (COllaborative Management Platform for detection and Analyses of [Re-] emerging and foodborne outbreaks in Europe) that involved both real samples (sewage and stool) and artificial in-silico generated datasets [17]; (ii) a ring trial in the Netherlands with the pilot phase based on eight clinical samples that assessed the variability in the sample preparation as well as the DNA and RNA extraction, the sequencing data then being analyzed by a single common bioinformatics workflow for all (IDbyDNA/UMCG, personal communication); and (iii) our study on the spiked plasma samples and artificial in-silico-generated datasets. 

These initiatives are pilot studies paving the way to standardizing quality controls in clinical viral metagenomics. In order to get these pilot studies into the production-level and ensure that participants can take advantage of these ring trials for their own laboratory accreditation, we find that it would be important to now consider implementing these ring trials in an ISO-accredited framework, notably by joining forces with the ISO17043 certified organizations currently running EQAs in clinical microbiology.

## Figures and Tables

**Figure 1 genes-10-00655-f001:**
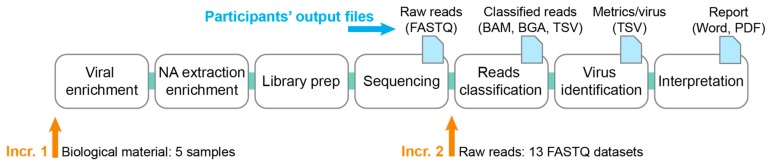
Viral metagenomics ring trial design. The figure shows the complete workflow from the sample to the report, which may include enrichment of viral particles, followed by nucleic acids (NA) extraction and viral nucleic acid enrichment, library preparation, sequencing, classification of reads, summary statistics per virus and final interpretation. For increment 1, five samples consisting of biological material were sent to participants (left orange arrow). For increment 2, participants received 13 FASTQ datasets (right orange arrow). Output files are listed at the top after each workflow step (blue arrow).

**Figure 2 genes-10-00655-f002:**
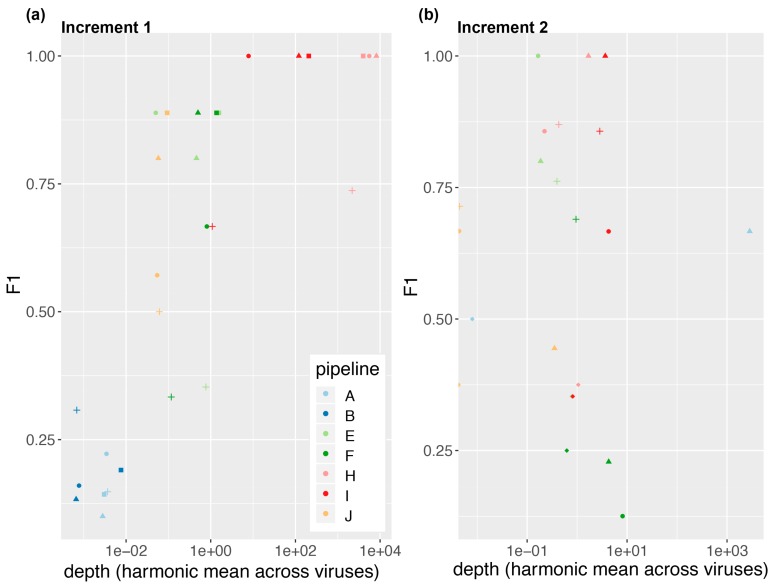
Performance as a function of depth (Equation (1)) for (**a**) increment 1 and (**b**) increment 2. We show here the results when participants used the SIB-provided common database. Markers represent the samples: Plasma_spike_1-1 (square); plasma_spike_1-10 (triangle); plasma_spike_1-100 (circle); NIBSC_multiplex (cross and diamond (note: In increment 2, NIBSC_multiplex was present in duplicate, cf. Materials and Methods).

**Figure 3 genes-10-00655-f003:**
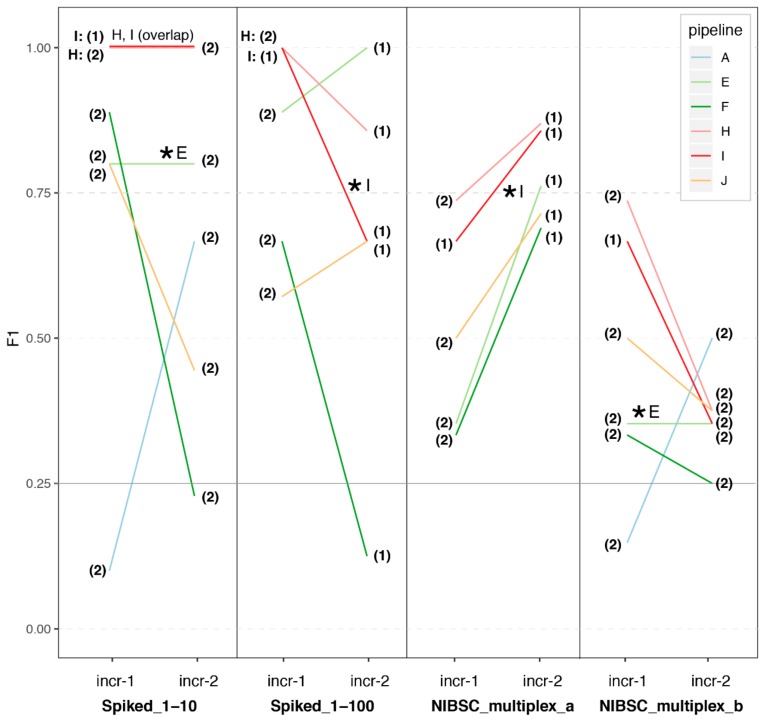
Performance across pipelines on the spiked biological samples (increments 1 and 2). We plot the F1-score obtained by each pipeline on the four spiked biological samples provided in increment 1. We also plot the F1-score obtained by these pipelines in increment 2, on the same samples but starting this time from one common FASTQ dataset that we selected from the FASTQ datasets submitted by participants of increment 1 (indicated by a “*”, see also Table 1 and Materials and Methods). We also indicate with a number if the analyses were based on the single-end (**1**) or paired-end (**2**) reads. Note that pipeline A did not report any results for two of these samples in increment 2 (Spiked_1-100; NIBSC_multiplex_a), and their results in the corresponding panels are therefore not represented here.

**Figure 4 genes-10-00655-f004:**
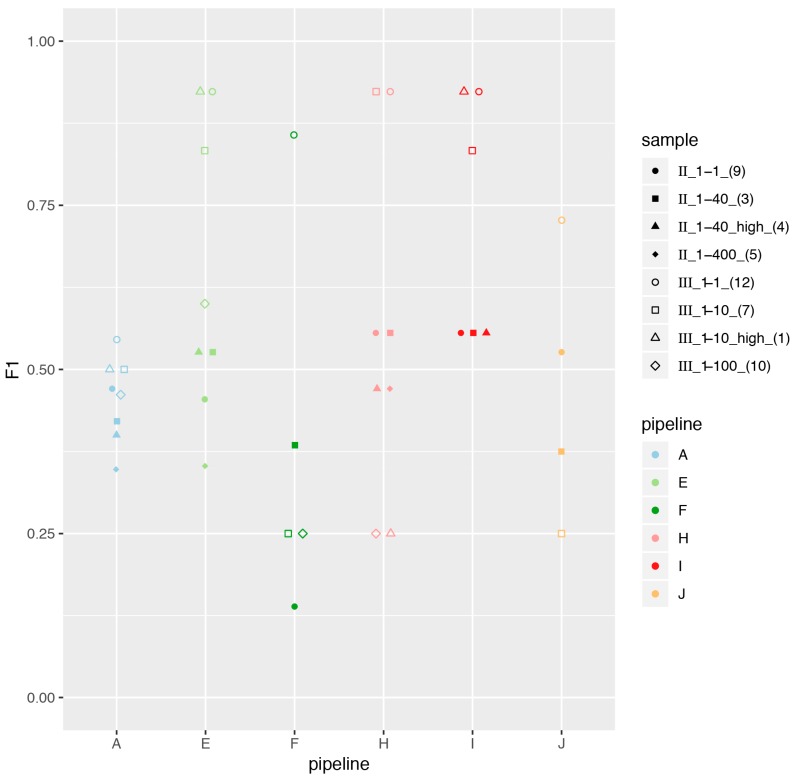
Performance across pipelines for increment 2 in-silico-generated datasets. We report the F1-score obtained by each pipeline on the artificial datasets, using the SIB-provided database. Note that participants sometimes reported no viruses at all in some of the artificial samples, and we could therefore not calculate an F1-score for these samples (missing markers in the plot). In addition, as pipeline I had identified viruses in various settings (1 × 100 bp, 1 × 150 bp, 1 × 250 bp), we display here the results obtained only with 1 × 150 bp, which correspond to the sequencing settings they used in increment 1. Note that data points were jittered horizontally for readability.

**Figure 5 genes-10-00655-f005:**
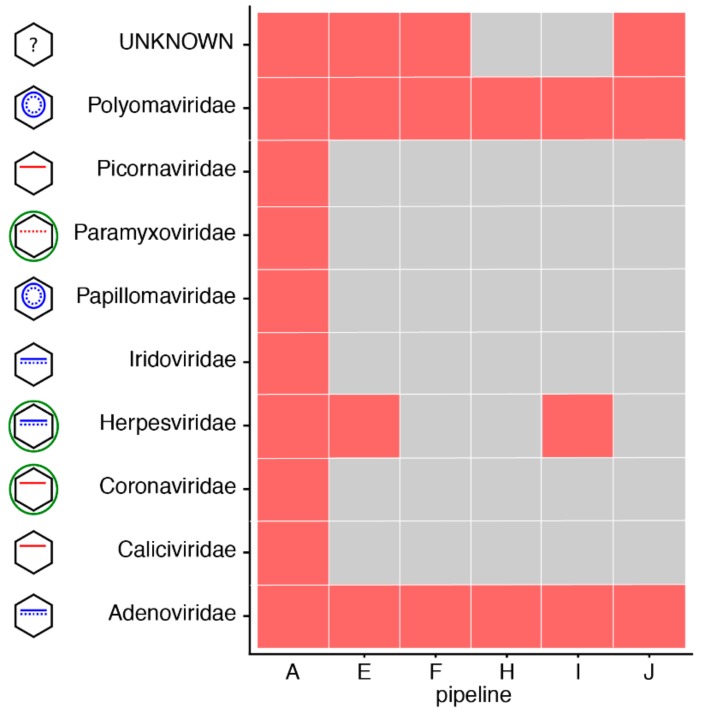
False positives identified by each pipeline in the negative sample of increment 2 (NIBSC negative control), when using the SIB common database. In order to compare results across the pipelines, we only show the family to which the false positive viruses identified by each pipeline belong (in red). The pictogram at the left of the family name gives an indication of the characteristics of the virus (envelope (green), DNA (blue; single or double stranded, linear or circular) or RNA (red, negative stranded if dotted line). Note that pipeline C is absent as it only returned results using its own in-house database, and not with the SIB common database.

**Figure 6 genes-10-00655-f006:**
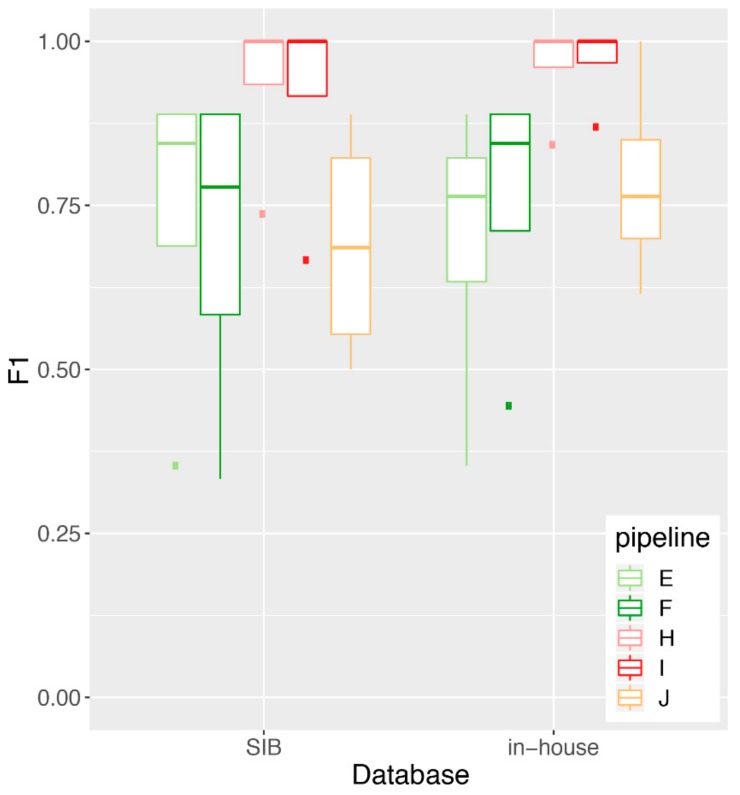
Impact of the database on performance for increment 1. We plot a boxplot of the performance (F1 score) achieved by each pipeline on the increment 1 samples, using either the SIB-provided common database or their own in-house database of reference sequences. No statistically significant differences in performance were observed between the SIB and the in-house databases (Mann-Whitney U test: *p* = 0.65, 0.95 confidence interval for difference in location [−0.54, 0.54] (pipeline E); *p* = 0.88 [−0.56, 0.44] (pipeline F); *p* = 1.0 [−0.26, 0.16] (pipeline H); *p* = 1.0 [−0.33, 0.13] (pipeline I); *p* = 0.56 [−0.5, 0.27] (pipeline J). The Mann-Whitney U test was performed in lieu of the Student’s t-test due to the fact that the F1 scores are not normally distributed, as verified by the Shapiro-Wilk test of normality (not shown). Note that pipelines A and B only reported results with the SIB common database and were therefore not included in this analysis.

**Table 1 genes-10-00655-t001:** Sample labels in increments 1 and 2.

Dataset Name	Label		Pipeline	Settings
	Incr-1	Incr-2	Incr-2	Incr-2
plasma_spike_1_1	5	NA	NA	NA
plasma_spike_1_10	2	11	E	2 × 150
plasma_spike_1_100	1	13	I	1 × 150
NIBSC_viral_control a	3	2	I	1 × 150
NIBSC_viral_control b	3	8	E	2 × 150
NIBSC_negative_control	4	6	A	2 × 150
II_1-1	NA	9	in-silico	various
II_1-40	NA	3	in-silico	various
II_1-40 high	NA	4	in-silico	various
II_1-400	NA	5	in-silico	various
III_1-1	NA	12	in-silico	various
III_1-10	NA	7	in-silico	various
III_1-10 high	NA	1	in-silico	various
III_1-100	NA	10	in-silico	various

NA: Not applicable; Incr: Increment.

**Table 2 genes-10-00655-t002:** General description of the in-silico generated datasets. The number (nb) of reads per virus was set here for the reads of length 150 bp. For the other read lengths, please refer to Appendix A.

	Dilution	Mutation Rate	Nb Reads/Virus
**II_1-1**	1:1	Normal	~2000/virus
**II_1-40**	1:40	Normal	~50/virus
**II_1-40_high**	1:40	high	~50/virus
**II_1-400**	1:400	normal	~6/virus
**III_1-1**	1:1	normal	~500/virus
**III_1-10**	1:10	normal	~50/virus
**III_1-10_high**	1:10	high	~50/virus
**III_1-100**	1:100	normal	~6/virus

**Table 3 genes-10-00655-t003:** Summary of the pipeline methodologies (tx: Treatment).

	A	B	C	E	F	J	I	H
Storage	−20 °C	−20 °C	NA	−20 °C	−20 °C	−20 °C	−20 °C	−20 °C
**Enrichment prior extraction**	None	None	NA	homogenization, centrifugation, filtration, nuclease tx			Centrifugation, filtration	Homogenization, centrifugation, DNase tx
**DNA extraction**	Trizol LS/Phasemaker	Trizol LS/Phasemaker	NA	QIAamp Viral RNA Mini Kit			easyMAG	easyMAG
**RNA extraction**	Trizol LS/Phasemaker	Not extracted	NA					Trizol
**Enrichment after extraction**	QuantiTect Whole Transcriptome Kit, Qiagen	None	NA	None	None	None	None	None
**Library preparation DNA**	Nextera XT DNA Library v2		NA	NEBNext® Ultra™ II DNA Library Prep Kit			Nextera XT	Nextera XT
**Library preparation RNA**			NA					TruSeq
**Sequencer**	MiSeq (2 × 150 bp)		NA	NextSeq (2 × 150 bp)			MiSeq (1 × 150 bp)	HiSeq 2500 & 4000 (2 × 100 bp)
**Reads filtering**	adapter trimming (bbduk.sh)			FastQC	Trimmomatic	Trimmomatic	PrinSeq, seqtk trimfq	Trimmo-matic
**Taxonomic classification**	k-mer		k-mer + mapping	mapping	mapping	k-mer	mapping	mapping
**Tool**	bbmap bbsplit		Kraken2 + Bowtie2	SeqMan NGen software v14 (DNAStar, Lasergene)	bowtie2	Kraken	bwa mem, blastn	SNAP

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
