# Peer review of "Viral Metagenomics in the Clinical Realm: Lessons Learned from a Swiss-Wide Ring Trial"

_genes, 2019, doi:10.3390/genes10090655_

Round 1

Reviewer 1 Report

In their manuscript entitled "Viral metagenomics in the clinical realm: lessons learned from a Swiss-wide ring trial", Dr. Junier et al. present methods and results of a national ring trial for metagenomics-based virus detection using mock samples and resulting sequencing data. This is an important contribution to this rapidly evolving field and a number of important performance determinants are addressed.

Comments

The authors state that "These results show no significant difference in performance for any given pipeline using the SIB vs. the in-house database (Mann-Whitney U test), suggesting that in our RT, it is the mapping/classification algorithm that is driving the overall performance, rather than the reference sequences database that is used along the algorithm." This result is unexpected in the broader context of diagnostic metagenomics and it is not clear how generalizable this finding is. At a minimum, this warrants more detailed discussion. It is not clear to this reviewer how much classification pipelines were re-optimized after inclusion of the common database. In general, interpretive criteria will depend on the size and compositon of the databases used. How were reporting criteria customized given the non-native database? The authors discuss the impact of human review and limited reproducibility of results with some pipelines. These are important limitations that should be discussed in light of the stated goal of accredited tests. This reviewer suggests to include specificity as an additional performance metric and split figure 6 into sensitivity and precision (or specificity). While the F1 score makes sense as a unifying measure, the diagnostic community is more familiar with sensitivity and specificity so these metrics should be provided, at a minimum in the supplement. The authors state thatv "these results suggest that sequencing should be deep enough to achieve 10X coverage of viruses, and that this is necessary but not sufficient for achieving high sensitivity and precision, as we still observe great variability of performance across different bioinformatics workflows." As discussed, requiring 10X coverage will substantially limit the analytical sensitivity of a metagenomics test for viruses, especially in samples other than plasma (i.e. higher biomass samples). For real-world use, pipelines have to be able to provide accurate results at coverage ranges well below 10X (and indeed often below 1X). The authors should discuss this limitation in more detail. Regarding negative controls: the authors state “Of these pipelines however, only pipeline H correctly identified that the sample was negative, thus being able to correctly deplete real contaminants from its data”. In real live, the utility of negative controls relies on them being representative of the samples being tested (regarding biomass etc); the authors should discuss how this may have affected the ability of pipelines E, F, H and J to correct for contaminants. The authors mention the important limitations of inconsistent viral classification schemes and the need to report viral names that are recognizable by the clinician audience. Given that the NCBI viral taxonomy does not consistently apply the species concept and many references are only partially annotated, the statement that the "taxonomic resolution of interest (species-level in our case) needs to be considered is correct but difficult to implement. The authors may want to discuss in more detail how this affects interpretation and comparison of results Figures in the supplement are not numbered Sequencing data should be submitted to the SRA or ENA 

Author Response

Dear Reviewer,

We would like to warmly thank you for your time and excellent comments for improving our manuscript on "Viral metagenomics in the clinical realm: lessons learned from a Swiss-wide ring trial". We reply below to all your comments.

The authors state that "These results show no significant difference in performance for any given pipeline using the SIB vs. the in-house database (Mann-Whitney U test), suggesting that in our RT, it is the mapping/classification algorithm that is driving the overall performance, rather than the reference sequences database that is used along the algorithm." This result is unexpected in the broader context of diagnostic metagenomics and it is not clear how generalizable this finding is. At a minimum, this warrants more detailed discussion.

We fully agree with the reviewer, this finding was also very unexpected to us. One reason for this finding may be that Swiss participating laboratories have hired bioinformaticians and dedicate resources in (semi-automatically/automatically) refining their internal databases, and do not simply download all NCBI viruses, likely achieving already good quality databases similar to the one we provided. We have updated our discussion in lines 580-595, where we also make sure to be cautious about this conclusion, since expert curated databases will certainly be useful for less sequenced viruses, as well as for increasing speed and reducing false positives calls in general.

Interestingly, we observed that the database that we provided enabled participants to get as good results as they would get using their own in-house database (Figure 6). One reason for this finding may be that participating laboratories have hired bioinformaticians and dedicate resources in (semi-automatically) refining their internal databases and do not simply download all NCBI viruses, likely achieving already good quality databases similar to the one we provided. Thus, in our settings (i.e. rather common viruses and Swiss clinical context), it appears that the impact of the database was rather low on overall performance. It is however unclear how generalizable this finding may be, given the limited number of tested viruses in our samples and the small number of participating laboratories.Since we did not test several databases of varying qualities, we would still recommend taking advantage of the expert curated databases that are being published (cf. Material and Methods and references therein) to optimize the speed of bioinformatics workflow, decrease false positive calls in general (cf. Figure S20), and also increase sensitivity for less sequenced viruses.

It is not clear to this reviewer how much classification pipelines were re-optimized after inclusion of the common database. In general, interpretive criteria will depend on the size and composition of the databases used. How were reporting criteria customized given the non-native database?

This is an important point. Participants did not re-optimize their pipeline after inclusion of the common database, as they were asked to use identical settings for a given pipeline in all the increments and when using the in-house or SIB-provided databases. We clarified this in the lines 270-272 as follows:

Participants were asked to use identical settings for a given pipeline in all the increments and when using the in-house or SIB-provided databases.

The authors discuss the impact of human review and limited reproducibility of results with some pipelines. These are important limitations that should be discussed in light of the stated goal of accredited tests.

This is indeed critical and we agree with you that this point should be stated more clearly in our manuscript. We updated the text in the Discussion in lines 625-635 as follows:

In this sense, we also suggest that the clinical viral community should share more interpretation and reporting practices, in order to further harmonize their practices and increase overall knowledge with more expertise sharing. The impact of human review and the observation that one pipeline did not reproduce its findings starting from an identical FASTQ dataset also calls for individual laboratories more clearly defining internal standard interpretation criteria, which would help reduce the impact of differential human review and likely be essential in an accredited context to ensure results reproducibility over time.

This reviewer suggests to include specificity as an additional performance metric and split figure 6 into sensitivity and precision (or specificity). While the F1 score makes sense as a unifying measure, the diagnostic community is more familiar with sensitivity and specificity so these metrics should be provided, at a minimum in the supplement.

We thank the reviewer for this suggestion. We actually used precision instead of specificity, as we did not have a means to assess the number of true negatives, thereby precluding the use of the specificity metric. We agree that the F1 score is a practical unifying measure, but that figures with both metrics separately should be provided at least in the supplement. We now provide these figures as supplemental Figures S20-21 and refer to them in the text as follows (lines 510-512):

We note that most of the F1-score signal was driven by differential performance in precision across pipelines, rather than sensitivity (Figures S20 and S21).

We also mention Figure S20 on differential precision across pipelines in the discussion about SIB vs. in-house databases.

The authors state that "these results suggest that sequencing should be deep enough to achieve 10X coverage of viruses, and that this is necessary but not sufficient for achieving high sensitivity and precision, as we still observe great variability of performance across different bioinformatics workflows." As discussed, requiring 10X coverage will substantially limit the analytical sensitivity of a metagenomics test for viruses, especially in samples other than plasma (i.e. higher biomass samples). For real-world use, pipelines have to be able to provide accurate results at coverage ranges well below 10X (and indeed often below 1X). The authors should discuss this limitation in more detail.

We thank the reviewer for this insightful comment. We moved part of this text to the Discussion and extended it as follows (lines 561-577):

Our ring trial was based on spiked plasma samples. Our results in Figure 2(a) suggest that in this context, a depth per virus of about 10X, as achieved by pipeline I, may be a rational target for high quality output results. Achieving a sequencing depth of 10X per reported virus can, however, be difficult and not always feasible, especially in samples other than plasma having higher biomassand for low replicating viruses. Thus, others have suggested to use instead >= 3 non-overlapping reads in distinct genomic regions as a criterion for reporting viruses in cerebrospinal fluid samples [1]. Ultimately, target depth of coverage might likely depend on the envisioned application and type of sample. This may also have consequences on what can be expected in terms of sensitivity and precision, and what sample processing steps, bioinformatics tools and interpretation criteria may need to be fine-tuned accordingly to compensate for e.g. lower depth of coverage per virus in some applications. As a result, performance criteria notably in an accredited setting may end up being application-specific to some extent, or require e.g. additional complementary validation tests to confirm the metagenomics results, as discussed in [3] for the identification of viral encephalitis, meningoencephalitis and meningitis. In the end, it is important to bear in mind that clinical metagenomics assays will likely not be used alone or as systematic first-line tests, but rather in combination with other diagnostic tests that all together will let clinicians take the best decisions for the patient.

Regarding negative controls: the authors state “Of these pipelines however, only pipeline H correctly identified that the sample was negative, thus being able to correctly deplete real contaminants from its data”. In real live, the utility of negative controls relies on them being representative of the samples being tested (regarding biomass etc); the authors should discuss how this may have affected the ability of pipelines E, F, H and J to correct for contaminants.

We thank the reviewer for this relevant comment. Indeed, participants of the ring trial did not have the possibility to use negative controls representatives of the samples being tested, since they lacked this information. We updated the Discussion as follows (lines 636-649):

Due to the design of the ring trial, participants of the ring trial did not have the possibility to use negative controls that were representative of the samples being tested, since they lacked this information. In increment 1, pipelines E, F, H and J nevertheless used internal controls alongside the provided samples (Figure S7). Pipeline H used a negative control (water submitted to the whole NGS process), whereas pipelines E, F and J used a well characterized sample as internal control (weak positive for one virus). Of these pipelines however, only pipeline H correctly identified the negative sample as being negative, thus being able to correctly deplete real contaminants from its data, while not adding type-I errors due to wrong mapping/classification in the bioinformatics step. Pipelines E, F and J however still reported viruses in the negative sample, suggesting these may arise mostly from type-I errors during the bioinformatics step. In increment 2 (Figure 5), we also observed that it is likely laboratory contaminants and type-I errors that affect participants’ false positive rate. In this sense, always running a negative control alongside is certainly very good practice to deplete laboratory contaminants, but not sufficient to avoid false positives calls in the bioinformatics step.

The authors mention the important limitations of inconsistent viral classification schemes and the need to report viral names that are recognizable by the clinician audience. Given that the NCBI viral taxonomy does not consistently apply the species concept and many references are only partially annotated, the statement that the "taxonomic resolution of interest (species-level in our case) needs to be considered is correct but difficult to implement. The authors may want to discuss in more detail how this affects interpretation and comparison of results.

This is indeed a very important point, notably to bear in mind for the future setup of similar ring trials. Having experienced this issue when analysing the results, we have now extended the discussion as follows (lines 549-558):

When establishing such ring trials, it is therefore important to clearly ask participants to report viruses at the most detailed resolution level possible, and provide them with means to report this (e.g. NCBI Taxonomy, or NCBI accession number). This is of particular importance since viral taxonomy is not consistently defined, and thus asking for virus identification may mean different taxonomic names and levels for different laboratories depending on the virus. The reported accession number will be useful to retrieve the virus full name, before asking if it belongs to the expected list of viruses at the chosen resolution level. In this sense, it is also essential that the list of allowed names for each of the expected viruses at the chosen resolution level is made available after such proficiency tests for participants review (cf. Table S1 in our ring trial).

Figures in the supplement are not numbered.

We have now numbered the figures in the supplement, thank you for noting this.

Sequencing data should be submitted to the SRA or ENA.

We also fully agree with the reviewer that all the data generated in this study should be available for others to also test and benchmark their pipelines. Thus, all the increment-2 FASTQ datasets (real and artificial) are published with the article as supplementary material and are available for download as a bundle on Zenodo. Whenever applicable (real sequencing data from increment 1), FASTQ headers have been anonymized.

We remain at your disposal should you have any further questions or comments.

Thank you very much and kind regards,

Dr. Aitana Lebrand

Reviewer 2 Report

Manuscript No.:            genes-573366

Title:                            Viral metagenomics in the clinical realm: lessons learned from a Swiss-wide ring trial

Authors:                       Junier et al

In this study, the authors evaluated the performance of five clinical laboratories, using seven different pipelines, to identify a series of viral genomes.  The manuscript describes the organization, running and outcomes from a preliminary ring trial implemented to benchmark sample processing/sequencing and bioinformatics pipelines for clinical viral metagenomics.  Although there is not a final consensus on “good” or “bad” approaches to samples preparation, deep sequencing and/or bioinformatics analysis, not surprisingly the authors conclude that the laboratories should aim to obtain as many reads (coverage) as possible when multiplexing samples in a single deep sequencing run.

Specific Comments:

Line 51: next generation sequencing (NGS) was already defined in line 47. Line 62: Change 'SIB Swiss Institute of Bioinformatics' to 'Swiss Institute of Bioinformatics(SIB)'. The abbreviation is then used on line 63. Table 3: Does pipeline 'I' have a method for RNA extraction? The box appears to be missing on the table. Lines 247 to 256: I understand the decision to reduce read numbers proportional to the read length, from the point of view of genome coverage; however, if this has a detrimental impact on longer-read pipelines insofar as the actual number of reads per target is lower. If virus detection and identification is purely based on coverage data, then there is no issue.  If mapped read numbers plays a role, could this cause a bias against the longer-read pipelines? Presumably this is accounted for in the depth measurement (mapped reads * read length / genome size). In preparing for this study did you run analyses (in any pipeline) using identical numbers of reads regardless of read length? If so, were the outcomes different to those reported for the normalized read numbers? Line 330: What does the (1) at the end of the line refer to? Line 338: it is not evident why pipelines A and B were based on the same FASTQ dataset, as the sample processing methods differ (lack of RNA extraction and post extraction enrichment in pipeline B)? Additionally, does 'For quantitect whole genome amplification of cDNA molecules' (Figure 3 pipeline A) refer to the QuantiTect whole transcriptome kit? The results section includes comment on the impact of sample processing (including potential points and levels of contamination) and bioinformatics pipelines (false positive and negative calling), however there doesn't seem to be much mention of these factors in the discussion.  Do the authors draw any overall conclusions regarding best practices for sample preparation/sequencing and/or details of bioinformatics pipelines? Or is the intent not to direct the practices of individual testing centers directly, but to define output quality guidelines regardless of the methods used to achieve report outcomes

Author Response

Dear Reviewer,

We would like to warmly thank you for your time and insightful comments in improving our manuscript on "Viral metagenomics in the clinical realm: lessons learned from a Swiss-wide ring trial". We reply below to all your comments.

Line 51: next generation sequencing (NGS) was already defined in line 47.

We updated the text accordingly, thank you.

Line 62: Change 'SIB Swiss Institute of Bioinformatics' to 'Swiss Institute of Bioinformatics(SIB)'. The abbreviation is then used on line 63.

We updated the text accordingly, thank you.

Table 3: Does pipeline 'I' have a method for RNA extraction? The box appears to be missing on the table.

Pipeline “I” used easyMAG for both DNA and RNA extraction. We clarified this in Table 3 by adding a missing cell border.

Lines 247 to 256: I understand the decision to reduce read numbers proportional to the read length, from the point of view of genome coverage; however, if this has a detrimental impact on longer-read pipelines insofar as the actual number of reads per target is lower. If virus detection and identification is purely based on coverage data, then there is no issue.  If mapped read numbers plays a role, could this cause a bias against the longer-read pipelines? Presumably this is accounted for in the depth measurement (mapped reads * read length / genome size). In preparing for this study did you run analyses (in any pipeline) using identical numbers of reads regardless of read length? If so, were the outcomes different to those reported for the normalized read numbers?

We thank you for this very interesting comment. No preparatory analyses were performed before the ring trial implementation and thus we do not have numbers on using identical numbers of reads for different read lengths. When preparing the datasets, we wanted to normalize in some way the number of reads to be fair between the participating laboratories. Indeed, having more reads is certainly an advantage for virus detection and identification, but that comes at a cost, resulting in a necessary trade-off to enable routine implementation in a clinical setting (we updated the text accordingly in lines 252-254).

However, in order to ensure fairness among participants, we decided to normalize the number of reads per virus to the read length, as having more reads is certainly an advantage for virus detection and identification, but comes at a cost, resulting in a necessary trade-off to enable routine implementation in a clinical setting.

As you mention, we also came to the conclusion that using coverage and average depth (measured as number of mapped reads * read length / genome size) as performance metrics would not ultimately penalize longer-reads pipelines. Also, as now shown in Table 3 that we updated, read length in our ring trial ranged from 100 to 150bp only. We had planned to also analyze artificial minION long read data (cf. Table S2 and Dataset S2), but unfortunately none of the participants actually chose to test those datasets.

Line 330: What does the (1) at the end of the line refer to?

The (1) refers to the formula number according to Genes Authors guidelines. We had however forgotten to cite it and now refer to it in the legend of Figure 2.

Line 338: it is not evident why pipelines A and B were based on the same FASTQ dataset, as the sample processing methods differ (lack of RNA extraction and post extraction enrichment in pipeline B)?

Indeed, the wording in the text is a mistake, A and B submitted different FASTQ datasets, although originating from the same sequencing center. Thank you again for the careful reading, we updated the text accordingly (lines 353-354):

Indeed, for the 7 pipelines, there were actually 5 sets of submitted FASTQ datasets from 4 sequencing centers, as pipelines E, F and J were based on a common FASTQ dataset.

Additionally, does 'For quantitect whole genome amplification of cDNA molecules' (Figure 3 pipeline A) refer to the QuantiTect whole transcriptome kit?

Indeed, the wording was not precise; we have updated Table 3 accordingly, thank you for noting this.

The results section includes comment on the impact of sample processing (including potential points and levels of contamination) and bioinformatics pipelines (false positive and negative calling), however there doesn't seem to be much mention of these factors in the discussion. Do the authors draw any overall conclusions regarding best practices for sample preparation/sequencing and/or details of bioinformatics pipelines? Or is the intent not to direct the practices of individual testing centers directly, but to define output quality guidelines regardless of the methods used to achieve report outcomes.

We very much thank the reviewer for this comment. When setting up the working group and implementing this trial, our idea was to view “NGS practices harmonization” as a means to achieve comparable high quality results, irrespective of the methodology chosen by each laboratory. Indeed, clinical laboratories need to integrate NGS as part of their existing practice, and may therefore have constrains e.g. on the choice of the nucleic acids extraction processes already in place for other existing analyses also requiring this step. We clarified this at the beginning of the Discussion as follows (lines 530-540):

“The SIB working group on “NGS Microbes Typing and Characterization” and the ring trial that we implemented aimed at harmonizing NGS practices in clinical viral metagenomics. Harmonizing means here for different clinical laboratories to achieve comparable results of high quality, irrespective of the methodology chosen at the experimental or bioinformatics levels, which may result in part from internal constrains as well (e.g. existing laboratory processes for nucleic acid extraction to be re-used as much as possible for metagenomics as well). Our results have highlighted various aspects that can impact overall performance at the experimental, databases and bioinformatics levels, offering insights to each participating laboratory into processes where they may further improve their workflows. In order for others to also benefit from this study and test their own workflows, we have published with this manuscript the database and all the datasets that were generated (Datasets S1 and S2).”

We remain at your disposal should you have any further questions or comments.

Thank you very much and kind regards,

Dr. Aitana Lebrand
